# Non-Vitamin K Antagonist Oral Anticoagulants in Patients with β-Thalassemia

**DOI:** 10.3390/biology12040491

**Published:** 2023-03-23

**Authors:** Michele Malagù, Filomena Longo, Federico Marchini, Paolo Sirugo, Andrea Capanni, Stefano Clò, Elisa Mari, Martina Culcasi, Matteo Bertini

**Affiliations:** 1Cardiology Unit, Azienda Ospedaliero-Universitaria di Ferrara, 44124 Cona, Italy; 2Day Hospital Thalassemia and Hemoglobinopathies, Azienda Ospedaliero-Universitaria di Ferrara, 44124 Cona, Italy

**Keywords:** thalassemia, hemoglobinopathy, anticoagulation, rivaroxaban, dabigatran, edoxaban, apixaban, direct, novel, atrial fibrillation, flutter, arrhythmias, thromboembolism, bleeding, prophylaxis

## Abstract

**Simple Summary:**

β-thalassemia is an inherited disorder characterized by the absent or reduced production of the β-chains of the hemoglobin. The natural history of the disease is radically changed after the introduction of iron chelation therapy, which prolongs life expectancy but also allows new comorbidities to become more prevalent. Among these comorbidities, atrial fibrillation is emerging as a clinical challenge since thalassemic patients show both higher ischemic and bleeding risk. Despite the worldwide adoption of NOACs in the general population, their use in thalassemic patients has been only described in case series and reports. The aim of the present study was to evaluate the efficacy and safety of NOACs in this cohort of patients. We enrolled 18 patients with transfusion-dependent β-thalassemia on treatment with NOACs for thromboembolic prophylaxis of supraventricular arrhythmias. During a mean follow-up duration of 22 ± 15 months, no thromboembolic events were reported, and no major bleedings were observed. The results of our study suggest the efficacy and safety of NOACs treatment in β-thalassemic patients. Given the increase in prevalence of atrial fibrillation in this cohort of patients, our results add an additional piece to the current state of knowledge and daylight practice.

**Abstract:**

**Background**. Patients with β-thalassemia have a high incidence of atrial fibrillation (AF) and other supraventricular arrhythmias. The use of non-vitamin K antagonist oral anticoagulants (NOACs) for thromboembolic prophylaxis in patients with β-thalassemia has not been systematically evaluated. **Methods.** We enrolled patients with transfusion-dependent β-thalassemia, who were on treatment with NOACs for thromboembolic prophylaxis of supraventricular arrhythmias. Data on thromboembolic and bleeding events were collected. **Results.** Eighteen patients were enrolled. The patients had a history of AF (sixteen), typical atrial flutter (five), and atypical atrial flutter (four). The patients were treated with dabigatran (seven), apixaban (five), rivaroxaban (four) or edoxaban (two). The mean follow-up duration was 22 ± 15 months. No thromboembolic events were reported. No major bleedings were observed. Three patients had non-major bleeding events. Two patients reported dyspepsia during treatment with dabigatran and were shifted to a different NOAC. **Conclusions.** Our study suggests the efficacy and safety of NOACs in patients affected by transfusion-dependent β-thalassemia.

## 1. Introduction

β-thalassemia is an inherited disorder characterized by the absent or reduced production of hemoglobin [1]. The disease is caused by mutations of the β-globin gene, transmitted with autosomal recessive inheritance, which leads to an imbalance in the synthesis of the globin chain [1]. The incidence is historically higher in the regions of the Mediterranean Sea, Middle East and Southern Asia, but in recent years the epidemiology of β-thalassemia has changed, especially as a consequence of migration, and the disease is now prevalent all over the world [2]. The disease is characterized by wide genotypic and phenotypic variability: the heterozygous state, which was previously considered a “minor” form of β-thalassemia, is usually asymptomatic and causes only mild anemia. On the other hand, homozygosis may result in β-thalassemia “intermedia”; which ranges in clinical severity from asymptomatic to transfusion-dependent forms, or in β-thalassemia “major”; which is the more severe disease, leading to a serious form of anemia in which survival is possible only with regular blood transfusions [3,4]. In the latter case, chelation therapy is required in order to reduce the adverse effects of iron overload due to red blood cell transfusions. Lifelong treatment with transfusions and iron chelation has changed the natural history of patients affected by β-thalassemia, allowing prolonged life expectancy [5]. The peculiar relationship between the underlying pathophysiology and the side effects of treatment lead to the development of several comorbidities, among which atrial fibrillation (AF) has emerged as a frequent issue, with a prevalence ranging from 2 to 33% [6,7,8,9]. In the general population, AF carries a thromboembolic risk and the patients affected may experience stroke, transient ischemic attack (TIA) or peripheral embolism [10]. On the other hand, thalassemic patients, even in the absence of AF, have a high incidence of thromboembolic events due to a state of chronic hypercoagulability and other clinical features such as therapeutic splenectomy [11,12]. Therefore, the presence of AF in patients with β-thalassemia determines a condition of marked risk.

Apixaban, dabigatran, edoxaban and rivaroxaban are non-vitamin K antagonist oral anticoagulants (NOACs). All of the four NOACs have been tested against warfarin (which was the previous “standard of care”) for the prevention of stroke and systemic embolism in patients with supraventricular arrhythmias, and have showed a better safety profile mainly due to the lower rate of bleeding events [13,14,15,16]. Therefore, NOACs are now recommended as the first choice drugs for the prevention of thromboembolic events in patients with AF and stroke risk factors, according to the CHA_2_DS_2_VASc score [17]. Furthermore, NOACs are also recommended in other arrhythmias, such as atrial flutter, and to prevent or treat pulmonary embolism (PE) and deep vein thrombosis (DVT) [17,18]. The clinical benefit of thromboembolic risk reduction from the treatment with anticoagulants always has to be weighed against a slight increase in bleeding events [19]. Considering the condition of patients suffering from β-thalassemia, the balance between thromboembolic and bleeding risk is of paramount importance.

Despite the worldwide adoption of NOACs in the general population, very little is reported in the literature regarding their use in patients with β-thalassemia [20]. The use of rivaroxaban only has been described in small case series and case reports [21,22]. To date, no study has described the use of apixaban, dabigatran or edoxaban in thalassemic patients.

## 2. Methods

Patients were enrolled and followed-up at the Azienda Ospedaliero-Universitaria di Ferrara, Italy. The inclusion criteria were: transfusion-dependent β-thalassemia, history of supraventricular arrhythmias (AF or atrial flutter), and treatment with NOACs for thromboembolic prophylaxis. The exclusion criteria were: age < 18 years, state of pregnancy, or inability to give informed consent.

Data regarding medical history, laboratory tests, ECG, echocardiography, magnetic resonance imaging and medical therapy were collected. Stroke, TIA and peripheral embolism were considered as thromboembolic events. Data on any kind of bleeding was collected. Adherence to treatment was specifically investigated.

This analysis was part of the project “*Atrial fibrillation in β-thalassemia*”, a monocentric study conducted at the Azienda Ospedaliero-Universitaria di Ferrara, Italy. The study protocol was approved by the Ethics committee and all patients signed informed consent. The study design was registered on *ClinicalTrials.gov* (ID: NCT05508932).

Categorical variables were expressed as a number and a percentage. Continuous variables were expressed as the median and interquartile range. Statistical analyses were performed with SPSS Statistics, version 25.0 (IBM, Armonk, NY, USA).

## 3. Results

A total of 18 patients were included in this study. All patients had transfusion-dependent β-thalassemia and were treated with NOACs. Eight were male (44.4%) and the median age was 55 years. All patients were treated with regular transfusions and chelation therapy. Thirteen patients received red blood cell transfusions every 14–15 days, while five patients received red blood cell transfusions every 21–22 days. Twelve patients were treated with iron chelation with deferoxamine, four patients with deferasirox and seven patients with deferiprone (three patients were treated with both deferoxamine and deferasirox, and two patients with both deferoxamine and deferiprone). Three patients had a previous history of stroke or TIA before treatment with NOACs. Baseline characteristics are showed in Table 1.

Sixteen patients had a history of AF while the remaining two had typical atrial flutter with no history of AF. Six patients had two or more documented supraventricular arrhythmias: two patients had both AF and typical atrial flutter, three patients had both AF and atypical atrial flutter, one patient had AF, typical atrial flutter and atypical atrial flutter. The mean CHA_2_DS_2_VASc score was 1.7 ± 1.7 (median 1, interquartile range 1–3).

Of the eighteen patients, seven were treated with dabigatran (six with the dosage 110 mg bid and one with 150 mg bid), five with apixaban (all with 5 mg bid), four with rivaroxaban (all with 20 mg once daily), and two with edoxaban (both with the reduced dosage of 30 mg once daily due to body weight < 60 kg, according to the dose reduction criteria [19]). The dose reduction criteria were applied when required, according to international recommendations [19].

Total “on treatment” time was 390 months, with a mean of 22 ± 15 months per patient (median 18 months, interquartile range 8–33).

No thromboembolic events were documented during the study period (Figure 1). In detail, in our cohort there were no strokes, no TIA and no peripheral embolism during treatment with NOACs.

Three patients had bleeding events. One patient had mild epistaxis while on dabigatran 110 mg, with no significant fall in hemoglobin levels and spontaneous resolution without NOAC interruption. One patient had hematuria secondary to nephrolithiasis while on apixaban 5 mg, with no significant fall in hemoglobin levels and spontaneous resolution without NOAC interruption. One patient, on rivaroxaban 20 mg, with previous history of urethrotomy and transurethral resection of the prostate, had recurrent episodes of hematuria during nephrolithiasis and after lithotripsy, requiring medical evaluation, with no significant fall in hemoglobin levels and spontaneous resolution without NOAC interruption. None of the patients required additional blood transfusions. None of the bleeding events was classified as a major bleeding, according to the criteria of the International Society on Thrombosis and Haemostasis (ISTH) [23].

Two patients experienced dyspepsia while taking dabigatran 110 mg bid. After shifting to a different NOAC (rivaroxaban 20 mg and apixaban 5 mg, respectively), the symptom resolved. None of the other patients stopped treatment. Two patients died during follow-up, one for refractory heart failure and one for respiratory failure. Those deaths were considered non-related to thromboembolic events.

## 4. Discussion

Our data suggest that NOACs may be effective and safe for thromboembolic prophylaxis in patients with β-thalassemia.

A small case series published in 2017 reported the use of rivaroxaban in eight patients with hemoglobinopathies, of whom four were affected by β-thalassemia, for AF or PE/DVT [21]. In that study, during a follow-up of 6–34 months, no patient experienced a thrombotic or bleeding event. Our results add evidence regarding the safety and efficacy of rivaroxaban and provide similar evidence with apixaban, dabigatran and edoxaban. Three of our patients had a previous history of stroke/TIA before treatment with NOACs. It is known that patients affected by hemoglobinopathies have a higher risk of thromboembolic events compared to the general population [11,12,24]. A previous study published in 2006, conducted before NOACs become available, reported a rate of thromboembolic events of 1.65% in a large cohort of patients with β-thalassemia major or intermedia [11]. Much has changed in the prognosis of thalassemic patients in the last 15 years. With the advancements in chelation therapy the prognosis of the disease has improved and survival has increased, and this has mainly been driven by a reduction of progressive heart failure. On the other hand, the prevalence of arrhythmic disorders has increased [6]. Therefore, the issue regarding chronic anticoagulant therapy in patients with β-thalassemia, who are fragile patients with different comorbidities, anemia and bleeding risk factors, became relevant and no longer anecdotal. The present study shed light on the use of NOACs in these patients. It is reasonable to consider that the absence of events “on treatment” in our cohort could be due to the small sample size and firm conclusions cannot be drawn. However, lacking larger contemporary studies, our results may be encouraging.

In our study cohort, three patients had bleeding events. Two were classified as non-clinically consequential minor bleedings and one was classified as clinically relevant non-major bleeding, according to the ISTH definition [25]. No major bleeding was observed. No patient required permanent discontinuation of oral anticoagulation. Previous studies reported a rate of bleeding (including any type of bleeding) of 14–18%/year in the general population [13,14,15,16]. In the case of β-thalassemia, bleeding events are of particular concern. However, our results seemed to indicate that NOACs may be safe in this population.

In our cohort, two patients had dyspepsia while taking dabigatran, which resolved after the shift to rivaroxaban and apixaban, respectively. Previous studies reported dyspepsia as the most frequent side effect of dabigatran, with an incidence of about 11% in the general population [16]. Our results confirmed that this side effect could also occur in patients with β-thalassemia. Whether the incidence of this side effect is similar or different in thalassemic patients compared to the general population is an open question. However, a shift to a different NOAC could be an option in those patients.

The CHA_2_DS_2_VASc score is a useful tool to stratify the risk of thromboembolic events in the general population and is generally accepted as a guide in the decision whether to initiate anticoagulation [17,26]. In detail, anticoagulant therapy is recommended (class I) in patients with CHA_2_DS_2_VASc ≥ 2 if male or ≥3 if female and should be considered (class IIa) in patients with CHA_2_DS_2_VASc = 1 if male or = 2 if female [17]. In the specific setting of patients with hemoglobinopathies, the CHA_2_DS_2_VASc score has not been validated [6]. Considering the peculiar condition of these kind of patients, it has been hypothesized that the CHA_2_DS_2_VASc score alone could not be enough and the expert consensus suggest to consider permanent anticoagulation at an early stage [27]. In our cohort, the mean CHA_2_DS_2_VASc was 1.7 ± 1.7 (median 1, interquartile range 1–3). However, some patients had criteria that did not match the CHA_2_DS_2_VASc scoring but were considered borderline or had additional risk factors (left atrial dilatation, smoking, impaired glucose tolerance) [17]. Furthermore, the CHA_2_DS_2_VASc ageing criteria is satisfied for age > 65 years but it is known that patients with thalassemia may have premature ageing compared to the general population [28,29]. These factors may explain the prescription of NOACs in our cohort. Further studies are needed to establish the criteria for the beginning of anticoagulant therapy in patients with β-thalassemia.

The use of deferasirox has been associated with an increased risk of gastrointestinal bleeding, especially in concomitance with anticoagulants [30]. In our cohort, no gastrointestinal bleeding was observed. However, this possible complication must be considered in this kind of patients and physicians should be alert for signs of gastrointestinal hemorrhage.

In the past, patients with AF, atrial flutter, PE or DVT were treated with vitamin K antagonist oral anticoagulants (i.e., warfarin). At present, due to a large body of evidence of the net efficacy and safety benefits of NOACs over warfarin, it is clearly recommended by international guidelines and consensus documents that, after the indication for oral anticoagulation is established, NOACs have to be preferred over vitamin K antagonists [17,19,31].

## 5. Limitations

Our study cohort was relatively small, being composed of 18 patients. A previously published cross-sectional case series reporting the use of NOACs in patients with hemoglobinopathies was composed of eight patients, of whom only four had thalassemia [21]. In the setting of sickle cell disease, a prospective study reporting the use of NOACs enrolled 12 patients [32]. In such a small body of evidence, we believe that our results could be sufficient to provide a contribution to the matter.

No thromboembolic events were documented “on treatment” in our cohort. It is reasonable to consider that the absence of events in our cohort could be due to the small sample size. However, lacking larger contemporary studies, our results suggest that NOACs treatment could be effective.

We presented a cohort study with no control group. Comparative studies could provide additional evidence.

## 6. Conclusions

In this cohort study among patients with β-thalassemia and a history of AF or atrial flutter, NOACs were effective in the prevention of thromboembolic events and safe.

## Figures and Tables

**Figure 1 biology-12-00491-f001:**
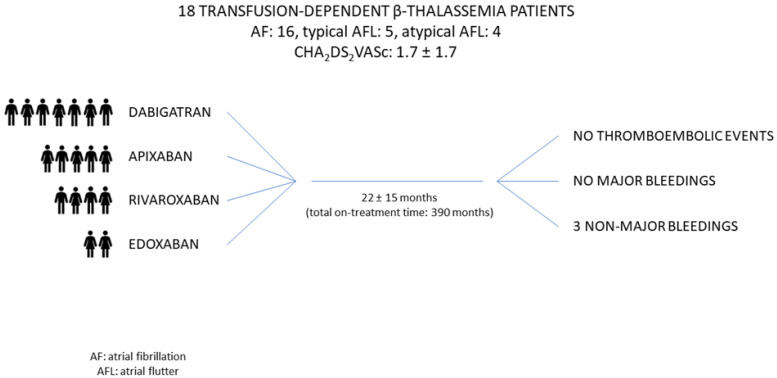
Non-vitamin K antagonist oral anticoagulants in patients with β-thalassemia.

**Table 1 biology-12-00491-t001:** **Patient characteristics.** Data are expressed as number (percentage) or median (interquartile range). COPD, chronic obstructive pulmonary disease; TIA, transient ischemic attack.

Variable	Study Population (*n* = 18)
Men	8 (44.4%)
Age (years)	55 (51–58)
Weight (kg)	56 (50–64)
Height (cm)	160 (152–169)
Body mass index (kg/m^2^)	22.0 (20.4–23.1)
Thalassemia major	17 (94.4%)
Thalassemia intermedia	1 (5.6%)
Hypertension	3 (16.7%)
Diabetes mellitus	5 (27.8%)
Impaired glucose tolerance	3 (16.7%)
Hypothyroidism	8 (44.4%)
Osteoporosis	12 (66.7%)
Splenectomy	17 (94.4%)
COPD	1 (5.6%)
History of heart failure	2 (11.1%)
Vascular disease	4 (22.2%)
History of stroke/TIA	3 (16.7%)
History of deep vein thrombosis	1 (5.6%)
Atrial fibrillation	16 (88.9%)
- Paroxysmal	12 (66.7%)
- Persistent	3 (16.7%)
- Permanent	1 (5.6%)
Atrial flutter	8 (44.4%)
- Typical	5 (27.8%)
- Atypical	4 (22.2%)
Pre-transfusion hemoglobin (g/dL)	10.5 (9.8–11.1)
Serum creatinine (mg/dL)	0.71 (0.55–1.10)
Ferritin (ng/mL)	294 (206–587)
Left ventricular ejection fraction (%)	60 (56–62)
Left atrial volume index (mL/m^2^)	46.1 (31.4–60.7)
T2* (ms)	37.5 (35–42)

## Data Availability

The data underlying this article will be shared on reasonable request to the corresponding author.

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
