# Peer review of "Non-Vitamin K Antagonist Oral Anticoagulants in Patients with β-Thalassemia"

_biology, 2023, doi:10.3390/biology12040491_

Round 1
Reviewer 1 Report
Dear authors,
this article investigate a fields of research in which there are few reported cases and it is thus interesting and novel. Nevertheless, the results presented are a collection of various and diffrent case studies since the treatments are different, the cohort of patients not homogeneous, the number of patients very low and there are not statistical conclusions. I would thus divide the results part in differnt parts, depending on the antagonist used and the patients treated, in order to make the study a proper collection of cases.
Some details are lacking in the description of the patients, in particular regarding the transfuion therapy used. Please introduce details about it for each patient and introduce this reference in the discussion (https://www.mdpi.com/2218-273X/11/11/1638) to discuss which therapy was used in your centre. Similarly, no details are reported for the chelation therapy used, please better sepcify it. If more exams are available for the patients follow up also introduce them (ECG, MRI, rtc..) since they can help to better describe the clinical cases.
Reviewer 2 Report
I find your work very significant, well thought out and structured, and regardless of the small number of subjects involved, it makes a significant contribution to haemostasis scientific field.
Reviewer 3 Report
1. it is a very small sample and the use of NOACs is very heterogenous
2.Authors have any previous data regarding treatment with vitamin K antagonist? can you provide a comparation? while NOACs are prefered not all settings have access to them and would be interesting to have a comparation between therapies in this particular kind of patients
Round 2
Reviewer 1 Report
The article reached the levels for publications in this journal although a larger population is necessary to validate these results.
Reviewer 3 Report
While the sample size is small, given the limited literature in the field it could serve as a base for larger investigations. It is valuable its publication at this time